# Road Surface Crack Detection Method Based on Conditional Generative Adversarial Networks

**DOI:** 10.3390/s21217405

**Published:** 2021-11-08

**Authors:** Anastasiia Kyslytsyna, Kewen Xia, Artem Kislitsyn, Isselmou Abd El Kader, Youxi Wu

**Affiliations:** 1School of Electronics and Information Engineering, Hebei University of Technology, Tianjin 300401, China; kwxia@hebut.edu.cn; 2Department of IT SBERX, Sberbank, 17997 Moscow, Russia; arcilite.kd@gmail.com; 3Department Biomedical Engineering, School of Electrical Engineering, Hebei University of Technology, Tianjin 300401, China; 201840000011@stu.hebut.edu.cn; 4School of Artificial Intelligence, Hebei University of Technology, Tianjin 300401, China; wuc@scse.hebut.edu.cn

**Keywords:** conditional generative adversarial networks, attention gate, road crack detection, dashboard images dataset

## Abstract

Constant monitoring of road surfaces helps to show the urgency of deterioration or problems in the road construction and to improve the safety level of the road surface. Conditional generative adversarial networks (cGAN) are a powerful tool to generate or transform the images used for crack detection. The advantage of this method is the highly accurate results in vector-based images, which are convenient for mathematical analysis of the detected cracks at a later time. However, images taken under established parameters are different from images in real-world contexts. Another potential problem of cGAN is that it is difficult to detect the shape of an object when the resulting accuracy is low, which can seriously affect any further mathematical analysis of the detected crack. To tackle this issue, this paper proposes a method called improved cGAN with attention gate (ICGA) for roadway surface crack detection. To obtain a more accurate shape of the detected target object, ICGA establishes a multi-level model with independent stages. In the first stage, everything except the road is treated as noise and removed from the image. These images are stored in a new dataset. In the second stage, ICGA determines the cracks. Therefore, ICGA focuses on the redistribution of cracks, not the auxiliary elements in the image. ICGA adds two attention gates to a U-net architecture and improves the segmentation capacities of the generator in pix2pix. Extensive experimental results on dashboard camera images of the Unsupervised Llamas dataset show that our method has better performance than other state-of-the-art methods.

## 1. Introduction

The conservation of city infrastructure requires the maintenance of road surfaces. Unproportioned usage of the roads in different parts of the city or rural areas influences the deterioration speed of a road surface. Cracks in the road surface are one of many warnings that roads require reconstruction [1]. The severity of this deterioration has a negative impact on the vehicles using it. Therefore, it is important to continuously receive information about the degree of deterioration. Constant monitoring is an ideal way to achieve automation in order to save money and human power.

Early detection of wear and tear on the roadway surface has a significant impact on the safety of traffic participants. The use of an instrument panel image data set helps to make the detection method applicable to drivers who are the primary victims of roadway surface deterioration. In traditional crack detection methods, the primary focus is on dealing with the problem by dealing with the various noise introduced by stains, spots, uneven illumination, blurring, and multiple scenes. Some methods assume that there is a clear distinction between noise and cracks in color tones, such as the iterative clipping method [2]. However, that method delivers a high false-positive rate when many dark spots and shadows as well as cracks occur in the test image. Some methods employ a shadow-removal algorithm to remove the road surface shadows while preserving the cracks, such as the CrackTree method [3]. This method has shown excellent precision and recall rates, but it is too complex and time-consuming. Some methods take both brightness and connectivity into account by measuring image texture anisotropy and produce good results on crack segmentation but are sensitive to edges, which may create a high false positive rate in some cases, such as in cycle texture anisotropy [4] and free-form anisotropy methods [5]. Zhang et al. [6] proposed the black top-hat transformation method and the threshold segmentation method to detect cracks in concrete tunnel surface images, but their method often fails in images with uneven illumination. Zhang et al. [7] established a multi-cue approach in the Aggregation Region and a creation area concept. However, such a technique was created primarily for small cracks within 2–5 pixels.

Yu et al. [8] introduced a multi-image-featured-based method for detecting cracks on concrete using an enhanced salp swarm algorithm applied to a support vector machine to improve the accuracy of crack detection. Additionally, Dong et al. [9] used encoder-decoder networks to perform pixel-level fatigue crack segmentation for steel structures. CrackForest [10] continues to identify cracks using handcrafted features, although they are insufficiently discriminative to distinguish fractures from complicated backgrounds with low level signals. Similarly, Li et al. [11] proposed a multiscale image function to detect cracks on roadway images.

Recently, Sobol et al. [12] and Park et al. [13] applied a variety of fully convolutional networks (FCNs) in order to detect cracks in pavement. These were trained in an end-to-end approach for pixel-level segmentation and detection. However, when processing some images, the network went into a state in which all pixels were processed as a background; a similar problem was also described in [14], where the method could not detect thin cracks. Oliveilra et al. [15] introduced a method that is based on support vector machines and applied it to pavement images where they were frequently broken into a succession of sub-depicted images during training. However, the method could not precisely segment out fracture curves throughout the entire image because the result is often only a label for the sub-image. Zou et al. [16] and Yang et al. [17] presented DCNNs for crack detection with hierarchical feature learning to identify pavement deterioration.

Ji et al. [18] proposed DeepLabv3+ to evaluate five parameters of the cracks. This method is an integrated approach based on the convolutional neural network for crack detection, as well as a crack quantification algorithm for crack quantification at the pixel level. Feng et al. [19] introduced a detection strategy based on the fusing of a deep convolutional neural network model. It combines the benefits of a single-shot multi-box multi-target detector with a U-net model for pavement crack identification. Their technology may offer pavement crack category information, accurate positioning, and geometric parameter information that can be used directly to evaluate pavement conditions. Haghighat et al. [20] investigated the use of deep learning models on intelligent transportation systems. They have emphasized the present use of deep learning approaches in problem areas that were previously addressed by analytical or statistical solutions.

The abovementioned methods have good results in the automatic detection of road surface cracks. However, these methods have a high rate of false-positive results on images taken in real-world. In other words, images captured by a camera in real-world settings without any post-processing. These images have several defects that increase false-positive results, such as:The roadway surface may be captured at degree angles other than the specifically established 90-degree angle, such as 55 degrees;There may be glares, stains, spots, and uneven illumination from the windscreen on some images;There may be shadow spots, uneven illumination, and other items of asphalt that can be detected as cracks on some images;There are also different white blanking techniques;Moreover, it may also include a variety of extraneous things that are unrelated to the roadway surface itself, such as sidewalks, road markings, cars, curbs, buildings, and locations where cracks may form.

All of these characteristics greatly reduce the efficacy of methods given in the related research. For example, CrackGAN [21] finds cracks throughout the image, despite the fact that detection of cracks on side-walks or the roadway edge consumes additional power capacity for ineffective difficulties. As a result, the selection measurement rate of CrackGAN is low. Furthermore, even though CrackGAN detects cracks correctly, the magnitude of the fractures may be incorrectly recognized. Since this method detects cracks as well as crack edges, it is no longer possible to use the observed crack shape for further crack investigation and measurement.

To solve this problem, we propose a method called improved conditional generative adversarial networks (cGAN) with attention gate (ICGA) for roadway surface cracks detection and developed a multi-level model. ICGA has two attention gates on a U-net architecture, which can improve the segmentation ability of the generator in cGAN. Therefore, ICGA can identify the targeted items more accurately. To obtain a more accurate shape of the detected target object, ICGA establishes a multi-level model with independent stages. In the first stage, everything except roads are treated as noise and removed from the image. These images are stored in a new dataset. In the second stage, ICGA determines the cracks. Therefore, ICGA focuses on the redistribution of cracks, not the auxiliary elements in the image.

## 2. Preliminary

### 2.1. cGAN

Generative adversarial networks (GAN) [22] are a form of neural network with a unique design that comprises a generator (G) that attempts to generate a new image and a discriminator (D). This architecture has proven to be a significant advancement in deep learning and has been used in processes such as style transfer, dataset generation, and many others.

Conditional GAN, or cGAN, introduced by [23] is an extension of the GAN architecture that allows for control over the image that is generated, such as enabling the generation of an image of a class [24]. During training, both the generator and discriminator are conditioned on some additional input. This could be any type of auxiliary data, such as a set of tags, a written description or class labels, or even data from different modalities.

The Pix2Pix method [25] is a general approach for image-to-image translation. It is based on the conditional generative adversarial network, where a target image is generated and conditioned on a given input image. In this case, the сGAN changes the loss function so that the generated image is plausible both in the content of the target domain and a translation of the input image. cGAN is frequently utilized in tasks such as image-to-image translation, image inpainting, and video prediction/generation. For example, it can be used to generate images with specific attributes from nothing but random noise, generate shadow maps, etc. This capacity of cGAN to generate images is the most interesting for us in this research. In this paper, we used Pix2Pix as the default architecure and modified it it for our purposes and improved it with attention gates.

### 2.2. Attention Gates

Attention gates (AGs) are a vital tool that has been demonstrated to be a game changer in deep learning. They are frequently utilized in tasks such as classification [26,27], machine translation [28], knowledge graphs, natural image analysis [29], and natural language processing (NLP) for image captioning [30]. Attention mechanisms have been used in deep learning, particularly in computer vision, to solve a wide range of problems, including image segmentation [31,32] action recognition [26], image classification [26,33], and image captioning [34]. The main principle of AG is shown as follows.

Let $x^{l}={\left\{x_{i}^{l}\right\}}_{i=1}^{n}$ be the activation map of a chosen layer $l\epsilon \left\{1,\ldots ,\;L\right\}$, where each $\;x_{i}^{l}$ represents the pixel-wise feature vector of length $F_{l}$ (i.e., the number of channels). For each $x_{i}^{l}$, AG computes coefficients $\alpha^{l}={\left\{\alpha_{i}^{l}\right\}}_{i=1}^{n}$ where $\alpha_{i}^{l}\in \left[0,1\right]$ to identify salient image regions and prune feature responses to preserve only the activations relevant to the specific task. The output of AG is ${\hat{x}}_{l}={\left\{\alpha_{i}^{l}x_{i}^{l}\right\}}_{i=1}^{n}$, where each feature vector is scaled by the corresponding attention coefficient.

## 3. Proposed Method

### 3.1. Parameter Selections

Attention coefficients, $a_{i}\in \;\left[0,1\right]$, are utilized to identify prominent visual regions and also to trim feature responses to retain only activations that are relevant to the task at hand.

The AGs output an element-wise multiplication of the input feature-maps and attention coefficients: x^i,cl=xi,cl×αil [35]. A single scalar attention value for each pixel vector xil∈ℝ Fl is computed, where Fl is the number of feature-maps in the *l*th layer whereas for multi semantic classes, multi-dimensional attention coefficients are used.

Multi-dimensional attention coefficients can be utilized to learn sentence embeddings according to [36]. In that paper, multi-dimensional attention coefficients can be learned, since numerous semantic classes are present in an image. As a result, each AG learns to concentrate on a subset of target structures. With multi-dimensional AGs, each $\alpha^{l}$ corresponds to a vector and yields ${\hat{x}}^{l}$, which can be calculated as follows.
(1)$${\hat{x}}^{l}=\left[\alpha_{\left(1\right)}^{l}⨀x^{l},..,\alpha_{\left(m\right)}^{l}⨀x^{l}\right],$$
where *α^l^_(k)_* is the *k*-th sub-AG and ⨀ is an element-wise multiplication operation. In each sub-AG, complementary information is extracted and fused to define the output of skip connection.

As a result, each AG learns to concentrate on a subset of target structures. For each pixel *i,* a gating vector gi∈ℝFg is utilized to define multiple focus zones. This gating vector comprises contextual information that is utilized to prune lower-level feature responses [35], which is employed for natural image categorization using AGs. The gating coefficient is obtained via additive attention, which has been proved experimentally to produce great precision. The following is a mathematical description of additive attention:(2)qattl=ψTσ1WxTxil+WgTgi+bg+bψ
(3)αil=σ2qattlxil,gi; Θatt,
where σ1x is an element-wise nonlinearity (e.g., rectified linear-unit) and σ2x is a normalization function. For our work, σ2xi,c=11 + exp−xi,c corresponds to the sigmoid activation function. AG is characterised by a set of parameters Θatt containing: linear transformations Wx∈ℝFl×Fint, Wg∈ℝFg×Fint, ψ ∈ℝFint×1, bias terms bψ ∈ ℝ, and bg ∈ℝ Fint.

For the input tensors, channel-wise 1 × 1 × 1 convolutions are used to compute the linear transformations [30]. As vector concatenation-based attention [35], the concatenated features *x* and *g* are linearly transferred to a ℝFint dimensional intermediate space. In this case, we picked a sigmoid activation function, which led to improved training convergence for all AG parameters. A grid-attention technique was also utilized, which collects information from various imaging scales for each skip link, increasing the grid-resolution of the query signal, and achieving improved performance.

### 3.2. ICGA

The overall architecture of our ICGA model is made up of a discriminator and a generator. The generator is a deep convolutional neural network that generates images, particularly conditional images, based on our improved U-Net algorithm that attaches attention gates. Both the original image from a dataset and its corresponding image with only manually identified cracks are used as input pairs, and the possibility that the crack occurs or does not exist in the selected image region is predicted by the discriminator.

#### 3.2.1. The Generator

The Pix2Pix architecture has been used in solving several image segmentation tasks, since they not only make effective use of GPU memory, but also provide excellent performance. Hence, we built our ICGA model on top of a standard Pix2Pix architecture. The generator of our ICGA model has the addition of attention gates. It samples or encodes the road image down to a bottleneck layer, then samples or decodes the bottleneck representation based on the output image. Coarse feature-maps are able to capture contextual information and thereby highlight the category and location of the foreground objects. These feature-maps are extracted at multiple scales and are later merged through the skip connections to combine coarse and fine level dense predictions. For our architecture, some skip-connections are inserted between the encoding layers and its corresponding decoding layers. Both the encoder and decoder of the generator are made up of standardized blocks of convolutional, batch normalization, dropout, and activation layers.

The AGs are included in the generator’s design to highlight critical characteristics via skip connections to accomplish our goal. Only relevant activations are combined prior to the concatenation procedure. During both the forward and reverse passes, AGs filter pixel activations. Gradients from background areas are less weighted during the backward pass. This allows for updating model parameters in shallow layers, which mostly depends on areas crucial to a certain work. The update rule for the convergence parameters of layer *l*-1 can be written as follows.
(4)$$\frac{\partial \left({\hat{x}}_{i}^{l}\right)}{\partial \left(\Phi^{l-1}\right)}=\frac{\partial \left(\alpha_{i}^{l}f\left(x_{i}^{l-1};\;\Phi^{l-1}\right)\right)}{\partial \left(\Phi^{l-1}\right)}=\alpha_{i}^{l}\frac{\partial \left(f\left(x_{i}^{l-1};\;\Phi^{l-1}\right)\right)}{\partial \left(\Phi^{l-1}\right)}\;+\;\frac{\partial \left(\alpha_{i}^{l}\right)}{\partial \left(\Phi^{l-1}\right)}x_{i}^{l}$$

Due to noise in the “real” photos, the network must provide major focus to the segmented part, such as the fractures on the roadway surface. This allows the network to spend less valuable computation and effort on irrelevant sections of an image and more on the parts that are cracks in order to correctly detect and specify the cracks that are present. Thus, we use attention gates in this case. The generator consists of 7-layers with 5 convolution layers and 2-layer attention gates. This careful configuration of the architecture has resulted in an increase in the accuracy of our network. The block diagram of the generator of our ICGA model is as shown in Figure 1.

#### 3.2.2. The Discriminator

The discriminator is a deep convolutional neural network that performs conditional image classification. by taking both the source image and the target image as input and predicts whether the target image is real or a fake translation of the source image. It’s architecture is built on Patch GAN [25]. Figure 2 depicts the discriminator of the ICGA model.

Discriminator receives 2 inputs: the input image and the target image that should be classified as real and the input image and the generated image (generator output), which should be classified as a fake. In other words, discriminator predicts an output as a probability in the range [0,1], which corresponds a real/fake prediction by the model.

#### 3.2.3. ICGA Model

The generator model takes an image pair as input and outputs a translated image version. The discriminator model is given input images of a real and generated image, and it is tasked with determining whether or not a fracture exists in the selected image area. The process continues over a number of training epochs with the generator learning to deceive the discriminator while minimizing the loss between the generated image and the target image. Figure 3 depicts the block diagram of ICGA architecture.

Below is the pseudocode ICGA model presented as Algorithm 1.
**Algorithm 1**: Modified Algorithm Training Loop Pseudocode1:Draw a minibatch of samples {X_AB_^(1)^,..., x_AB_^(m)^} from domain X2:Draw a minibatch of samples {Y^(1)^,..., y^(m)^} from domain Y3:Compute the discriminator loss on real images:IrealD=1m∑i=1mDAAi−12 + 1n∑i=1nDBBj−124:Compute the discriminator loss on fake images:    IfakeD=1m∑i=1mDBGA→BAi 2 + 1n∑j=1nDAGB→ABj 25:Update the PatchGAN discriminator6:Apply Attention to the Generator        qattl=ψTσ1WxTxil+WgTgi+bg+bψ)            αil=σ2qattlxil,gi; Θatt7:Compute the B → A generator loss:         JGB→A=1n∑j=1nDAGB→ABj−1)2+JB→A8:Compute the A → B generator loss:         JGA→B=1m∑i=1mDBGA→BAi−1)2+JA→B9:Update the generator

Having two attention gates enables our network to effectively estimate the detection rate of cracks in Figure 4. As the time increases (epochs 180, 186, 192, 195, 230, 383, and 447), the model significantly improves at detecting cracks, as illustrated in the image. However, the ICGA approach detects cracks across the entire given image, even in places that are irrelevant for our purpose, such as sidewalks, and incorrectly recognizes road boundaries as cracks. Thus, we have not yet reached our goal of identifying only the cracks in the roadway surface. As a result, more work on this study is required.

### 3.3. Method Steps for the Road Surface Crack Detection

Instead of recognizing all the cracks in an image, we want to design a tool that detects cracks only on the roadway surface. Therefore, we do not need to detect cracks on the sidewalk or recognize road margins as cracks. Furthermore, when using genuine photographs from dashboard cameras, we must note that such images contain high-quality yet live images. The photographs used in the dataset are not captured at a camera angle of 90 degrees to the roadway surface. Images have glares from the windscreen, various white-balance, and other “defects”, which can also cause incorrect crack detections. Hence, the stages of the procedure are as follows.

(1)The first stage entails preparing the dashboard image dataset.(2)The second stage is road segmentation, which involves removing extraneous items from images using the ICGA method. We train our model to remove unwanted items from the images, such as scenery, construction, walkways, etc., as described in Section 3.2. As a result, we obtain a new dataset, which we called Roadway cracks.(3)The third stage is image pre-processing for the processing of the new Roadway cracks dataset. It contains image resolution transformation and channel configuration.(4)In the fourth stage, we apply the ICGA method to detect cracks on the newly segmented dataset.

The recurrent usage from the first to the fourth phases improves the accuracy of crack detection not only due to the ICGA method, but also due to image pre-processing. Figure 5 shows the implementation scheme of our method.

## 4. Results

### 4.1. Experiment Preparation

#### 4.1.1. Dataset Description

To validate the performance, the Unsupervised Llamas dataset [37] was selected. This dataset contains 100,000 photos that have been tagged and have a resolution of 1276 × 717 pixels. This dataset’s photos are taken from approximately 350 km of recorded driving. Dashboard black-and-white camera images, pixel-level annotations of dashed lane markers, 2D and 3D endpoints for each marker, and lane associations to link markers are all included in the collection. This dataset allows for a clear comparison of various detection algorithms.

The roadway surface in the dataset images is captured the same way as the dashboard camera sees it. As a result, the dataset images have the following “defects”, which are followed by “live” photographs. The image captures a 55-degree angle view of the roadway surface. Some images feature glares, stains, spots, and uneven illumination from the windscreen. Some images have shadow spots, uneven illumination, and other asphalt imperfections that are not cracks. Some images differ in white-balance, since they were captured in cloudy or sunny weather, and there are other objects in the image that are not related to the roadway surface itself, such as sidewalks, road markings, cars, curbs, and buildings.

We use two datasets manually constructed from this data to investigate the applicability of ICGA.

The first dataset consists of a set of photographs from the Unsupervised Llamas dataset [21] and the same images without any additional items other than the roadway (henceforth referred to as the “Dashboard image dataset”). We manually removed superfluous items from the dataset images in order to create a dataset of examples for training our network that excluded unneeded objects. There are 405 training images and 105 test images in the Dashboard image dataset.

The second dataset consists of the same set of images that we used in the first dataset, as well as images with manually discovered cracks. The difference is that these images are a result of segmenting only the roadway surfaces in the images using our ICGA model. Unnecessary objects in the dataset photos are automatically removed while filling the rest of the image with white pixels. This new dataset is hereby referred to as the “Roadway cracks dataset”. The number of training images is 405, while the number of test images is 105. Figure 6a,b depicts Dashboard image and Roadway cracks dataset samples, respectively.

#### 4.1.2. Experimental Environment

For our experiment we used a system with the following configuration:Linux based system: Ubuntu 20.04;PyTorch platform with CUDA based video cards 4X 1080 TI;A GPU video memory of 11 Gb;CPU: Intel(R) Xeon(R) Silver 4114 CPU @ 2.2 Hz;Server model: DELL PowerEdge T640 tower server;32 GB memory, 10 TB hard drive, and 320 GB solid state drive;The programming software was Python 3.

Training details are the following. The training method was fairly typical with that of the image-to-image conversion task. The Adam optimizer was a generic version of gradient descent to make training more stable and efficient. The learning rate was set to between 0.002 and 0.001. The batch size was set to 1.

Speed and data requirements are the following. The dashboard images are made up of only 405 training images and 105 test images. Thus, the images of road cracks are limited to 405 training shots and 105 test images. Training can be swift on datasets of this size. For example, the findings displayed in Figure 7 after 400 epochs required six hours of training on a single NVIDIA GeForce GTX 108 GPU.

### 4.2. Evaluation Metric

To evaluate the effectiveness of the proposed technique, various competitive methods were selected, such as CrackIT [15], CrackForest [10], FCN-VGG [14], DeepCrack-1 [16], DeepCrack-2 [16], Pix2pix GAN [25], and CrackGAN [21]. For accurate comparison with our results, we ran experiments using the same equipment and platform environment, as well as both the Dashboard image dataset and the Roadway fractures dataset. We employed evaluation criteria such as positive predictive value (PPV) [37], true positive rate (TPR), F1, and Hausdorff distance score (HD-score) [30,38] to assess the accuracy of crack detection, since many related researchers employed these criteria to evaluate the performance of the proposed algorithms [1,3,9,10,11,15,16,19,21]. Moreover, we will employ more criteria to evaluate the performance of our method in future studies.

*PPV* quantifies the proportions of positive and negative outcomes in the Roadway cracks dataset that are correctly or incorrectly identified as cracks.
(5)$$PPV=\frac{P\_tp}{P\_tp+P\_fp}$$

*TPR* (or recall or sensitivity) represents the fraction of image segments with cracks found in the total number of images in the Roadway cracks dataset. In other words, this measures how successfully our classifier detects items in the image (i.e., defines cracks).
(6)$$TRP=\frac{P\_tp}{P\_tp+P\_fn}$$

The *F-score* is a performance metric that shows the correctness of a test. It is derived from the test’s precision and recall. The maximum number for an F-score is one, showing flawless precision and recall, while the lowest potential value is zero, indicating that either the precision or the recall is zero.
(7)$$Fscore=\frac{2\times PPV\times TPR}{P\_tp+P\_fn}$$

The HD-score represents overall crack localization accuracy and is unaffected by the foreground–background imbalance inherent in long-narrow object identification.

### 4.3. Experiments on Road Segmentation and Cracks Detection

This experiment comprises two stages: road segmentation and cracks detection. Table 1 and Figure 7 show the results of segmenting the original Unsupervised Llamas dataset using both pix2pix cGAN and our proposed model ICGA.

Figure 7 shows that the ICGA method has better performance than all competitive models. The reason is that our method removes unnecessary objects from the images, leaving only the roadway surface in the image. Hence, the results of our method are better than those of all competitive models.

Table 1 shows that our model outperformed the Pix2pix cGAN model. For example, the F1 of Pix2pix cGAN is 91.07%, while that of our model is 92.01%. This is due to the addition of attention gates to the modified U-Net, which served as the foundation of the generator in the ICGA overall cGAN model. Therefore, the introduction of AG is beneficial to the model. Hence, our model has better performance than the Pix2pix cGAN model.

cGAN is our primary solution for crack detection. It enables us to train the neural network to filter the roadway surface, remove extraneous picture objects, and leave just the parts recognized as cracks on the images. To verify the effectiveness of our strategy, six competitive methods, CrackForest, FCN-VGG, DeepCrack-1, DeepCrack-2, Pix2pix cGAN, and CrackGAN were run on the Roadway cracks dataset. The comparison of quantitative evaluations on Roadway cracks dataset is shown in Table 2.

Table 2 clearly shows that our model has better performance than all the competitive models. For example, the F1 score of our model outperforms the closest model to it by up to 2.0% and with a TPR gap of up to 11%. The details are shown as follows.

CrackForest finds cracks with 23.21% accuracy, and these findings are insufficient for running an automated program. If you look closely at the output images, you will notice that there is a lot of noise in the end result. It correlates to the identified road background grain, confirming the method’s unsuitability for our purpose due to its high sensitivity.

Cracks are detected with 0% accuracy by FCN-VGG. There are no cracks observed in samples 1 and 3, as shown in Figure 7. As a result, when the fractures are narrow, the rate of false negatives is considerable. It can be explained due to higher complexity of input images used in this paper as images captured by a camera in real-world settings without any post-processing.

Moreover, Table 2 provides the comparison of the results of other methods on the Roadway cracks dataset. The result of FCN-VGG and DeepCrack-2 on the Roadway cracks dataset is “All Black” images and value “0”. A similar situation could be found in other papers [9,21], We used images captured by a camera in real-world settings without any post-processing. This means, that our input images may have glares, stains, spots, and uneven illumination from the windscreen on some images; images can also have different white blanking techniques. In addition, the crack itself has a complicated shape; its edges may be uneven and in different colors. As a result, FCN-VGG, by using the pixel recognition method, may not always positively define a cluster of pixels as cracks. DeepCrack–2 detects the background of every crack and non-crack patch and generates photos that are completely black.

DeepCrack–1 and CrackGAN techniques recognize all cracks; however, the size of the edges of the detected cracks does not match the groundtruth. In other words, while these approaches detect cracks, the results are insufficient for performing crack measurements for future reference. As a result, DeepCrack–1 has a 46.03% accuracy and CrackGAN has a 79.34% accuracy. Furthermore, these methods produce an excessively high number of erroneous findings. The training photos contain a large number of extraneous items that are not cracks.

The Pix2pix cGAN approach yields high numerical results (73.04% accuracy). However, the geometry of the identified cracks does not match the shape of the cracks on the pavement, as seen in Figure 7. Furthermore, Pix2pix cGAN displays various typical failure instances. For example, thin asphalt shadows are defined as cracks, whereas wide shadows are not defined as cracks.

Our ICGA model achieves a staggering 88.03% accuracy, which is approximetly a 9% gain over the previous best model, the CrackGAN model. The result shows how efficient the images our method generates are, which are quite close to the groundtruth.

## 5. Conclusions

Overall, changes to assist traffic management and traffic planning will improve transportation road safety and security, reduce maintenance costs, optimize transportation performance for both public and ride-sharing firms, and promote driverless car development to a new stage. Therefore, the importance of a reliable roadway crack detecting system cannot be overstated.

To effectively detect cracks in the images taken in real-world contexts, in this study, we proposed the ICGA with multimodel learning based on an enhanced cGAN algorithm that makes use of attention gates on the U-net architecture for fracture identification on roadway surfaces. To obtain a more accurate shape of the detected target object, ICGA establishes a multi-level model with independent stages. In the first stage, everything except roads is treated as noise and removed from the image. In the second stage, ICGA focuses on the redistribution of cracks, not the auxiliary elements in the image. ICGA adds two attention gates to a U-net architecture and improves the segmentation capacities of the generator in Pix2Pix. Experimental results show that our method has better performance than other state-of-the-art methods.

## Figures and Tables

**Figure 1 sensors-21-07405-f001:**
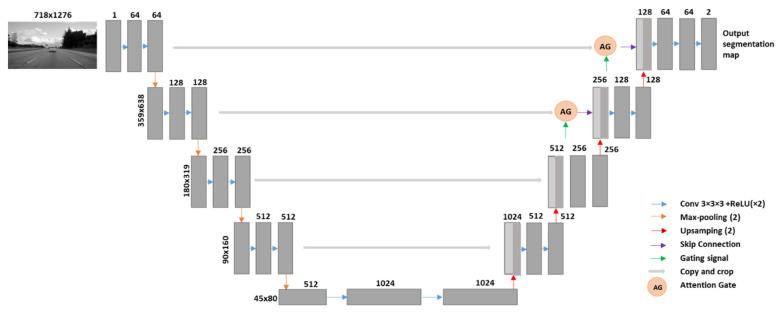
A block diagram of the genertor of our proposed ICGA model.

**Figure 2 sensors-21-07405-f002:**
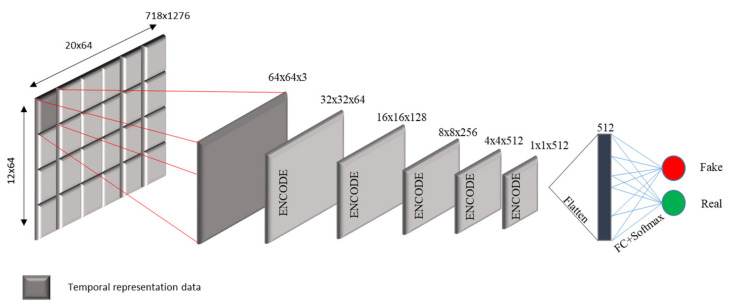
Architecture of the PatchGAN Discriminator of ICGA model.

**Figure 3 sensors-21-07405-f003:**
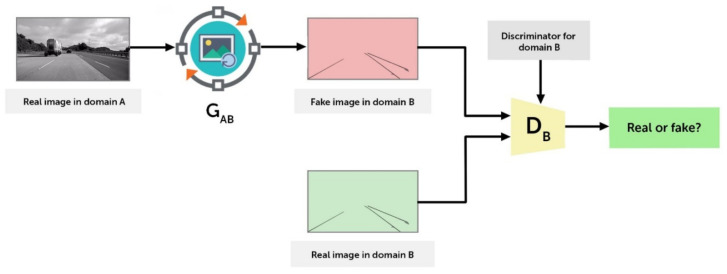
Block diagram of ICGA architecture.

**Figure 4 sensors-21-07405-f004:**
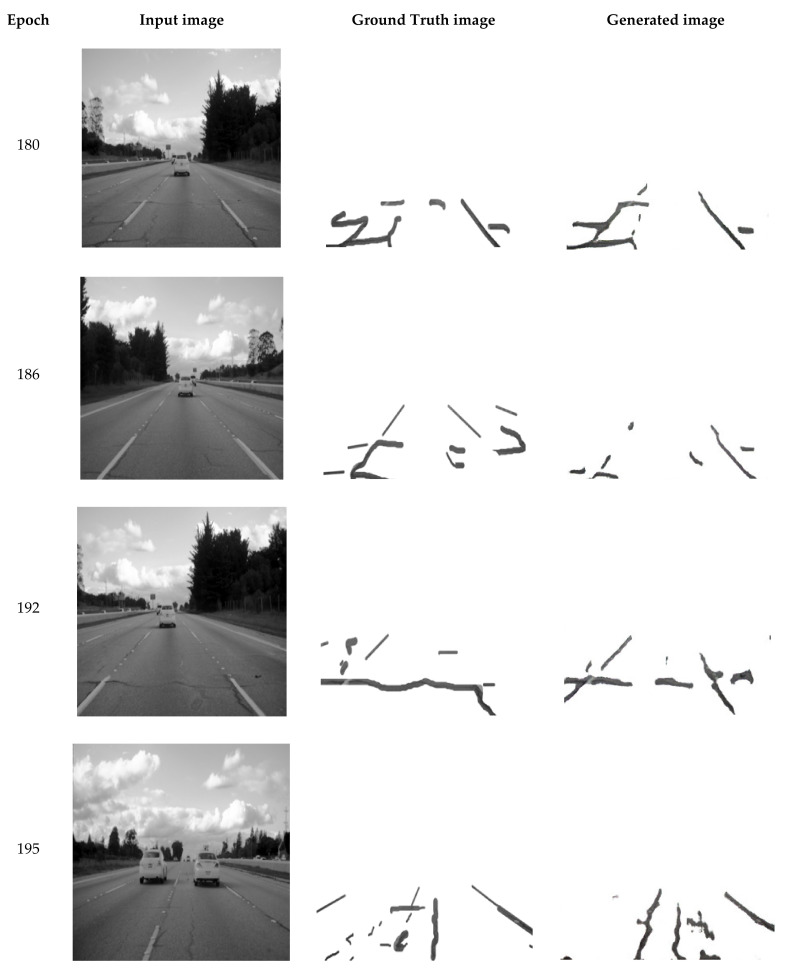
Images showing the results of our improved model.

**Figure 5 sensors-21-07405-f005:**
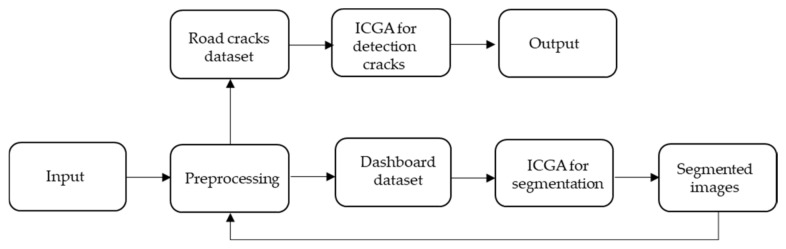
Implementation scheme of our multi-level model.

**Figure 6 sensors-21-07405-f006:**
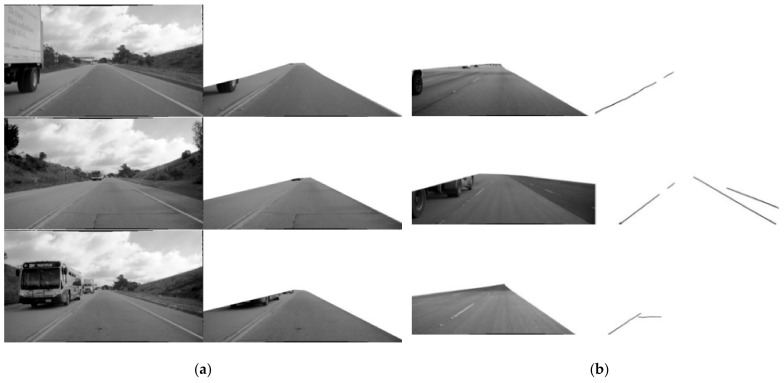
Examples of images datasets: (**a**) road without landscape, construction, sidewalks, and other objects; (**b**) road sections classified as cracks.

**Figure 7 sensors-21-07405-f007:**
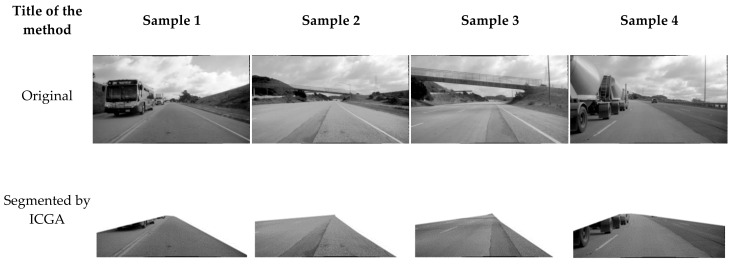
Comparison of results of segmentation and cracks detection for the Roadway cracks dataset.

**Table 1 sensors-21-07405-t001:** Quantitative evaluations on the Dashboard image dataset.

Methods	PPV	TPR	F1	HD-Score
Pix2pix cGAN	89.24%	95.04%	91.07%	92
ICGA	89.33%	95.06%	92.01%	94

**Table 2 sensors-21-07405-t002:** Comparison of quantitative evaluations on the Roadway cracks dataset.

Methods	PPV	TPR	F1	HD-Score
CrackForest	23.21%	77.03%	5.07%	47
FCN-VGG	0.00%	0.00%	N/A	N/A
DeepCrack-1	46.03%	76.04%	44.32	53
DeepCrack-2	0.00%	0.00%	N/A	N/A
Pix2pix cGAN	73.04%	79.07%	83.01%	82
CrackGAN	79.34%	72.27%	76.31	67
ICGA	88.03%	90.06%	85.01%	94

## Data Availability

Not applicable.

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
