# Peer review of "Road Surface Crack Detection Method Based on Conditional Generative Adversarial Networks"

_sensors, 2021, doi:10.3390/s21217405_

Round 1
Reviewer 1 Report
Reviewers Comments on Road Surface Crack Detection Method Based on Conditional Generative Adversarial Networks
Comments:
- The paper presents an Improved Generative Adversarial Networks with Attention Gate (ICGA) for roadway surface cracks detection. Though, an attempt was made to describe the concept of the proposed ICGA, Unet architecture and the incorporation of the attention gate. The manuscript could benefit from several improvements despite different contradicting statement and a neutral contribution to knowledge
- The last sentence in section 2.1 “In this paper, we used Pix2Pix as the default architecture and made a modification of it for our purposes and improved it with Attention Gates” and the first sentence in 3.2 “In this work, we build our ICGA model based on a cGAN architecture. The generator of the ICGA model is built on top of a standard 2D U-Net [31] architecture with addition of addition gates”. {Though the second addition ought to be attention gate} are contradicting each other. Is the proposed ICGA based on pix2pix or U-net?
- Authors need to clarify the function of or used of the discriminator as the only place it was mention in the manuscript was in “The architecture of the discriminator is built on Patch GAN” and no further details
- The structure of the last paragraph on page 5 gives the impression that the U-Net is responsible for the classification of whether an image is real or fake. However, it was described as a generator. Shouldn't that be the work of the discriminator?
- What is the novelty in the used of attention gate in the U-Net architecture, as several documented literature such as:
- Khanh, T. L. B., Dao, D. P., Ho, N. H., Yang, H. J., Baek, E. T., Lee, G., ... & Yoo, S. B. (2020). Enhancing u-net with spatial-channel attention gate for abnormal tissue segmentation in medical imaging. Applied Sciences, 10(17), 5729.
- Huang, Z., Zhao, Y., Liu, Y., & Song, G. (2021). GCAUNet: A group cross-channel attention residual UNet for slice based brain tumor segmentation. Biomedical Signal Processing and Control, 70, 102958.
Among others have proposed and apply the techniques for image related segmentation and detection as well as the used of the pix2pix approach.
Reviewer 2 Report
- the functions of (1) and (2) are described unclearly.
-
You have describe two "element-wise multiplication operation" in function (3) and page4-lines4, what's the meannings and differences between them ?
-
the work pattern of AG is unclearly illustrated, and must be rewritten by a sub-diagram.
-
Fig.4 should be moved to experiment section.
-
Choose the general standard evaluation metric functions instead of your's.
-
Table2, it's unreasonable for FCN-VGG and DeepCrack-2.

Reviewer 3 Report
A method commonly used to generate or transform the images for cracks detection on the road surface is the Conditional Generative Adversarial Networks (cGAN).
In this paper a method called improved cGAN with attention gate (ICGA) for roadway surface cracks detection, is used to obtain a more accurate shape of the detected target object.
ICGA establishes a multi-level model with independent stages.
The manuscript is interesting, fits well with the aim of the Sensors Journal.
The reviewer recommends improvement in the following items:
- In the Introduction, The Authors state that "To address the challenge of accurately detecting and quantifying cracks, various methods have been proposed [8-18]." The Authors should briefly explain what these methods are and how they differ from each other.
- In the Introduction there is a list of defects of images captured by a camera in real-world settings without any post-processing. These defects could be listed using a bulleted list, and not written inline in the text.
- After Equation (3) the sentence starts with a comma. This is an error; the comma should be inserted at the end of the equation (or omitted).
- Seven lines after Paragraph 4.2, please leave a space between " (HD-score)" and " [30],[37] ". Please check the entire manuscript for similar typos.
According to what said above, the reviewer’s opinion is that the manuscript can be accepted for publication after the described minor revisions.
Round 2
Reviewer 2 Report
More details should be illustrated in:
Figure 3. Architecture of the PatchGAN Discriminator of ICGA model.
